# Immunolocalization of Jasmonates and Auxins in Pea Roots in Connection with Inhibition of Root Growth under Salinity Conditions

**DOI:** 10.3390/ijms242015148

**Published:** 2023-10-13

**Authors:** Guzel Akhiyarova, Gyulnar Vafina, Dmitriy Veselov, Guzel Kudoyarova

**Affiliations:** Ufa Institute of Biology, Ufa Federal Research Centre, Russian Academy of Sciences, Prospekt Oktyabrya 69, 450054 Ufa, Russia; akhiyarova@rambler.ru (G.A.); vafinagh@mail.ru (G.V.); veselov@anrb.ru (D.V.)

**Keywords:** auxin, immunolocalization, jasmonic acid, root length, root meristematic zone, *Pisum sativum* L.

## Abstract

Inhibition of root elongation is an important growth response to salinity, which is thought to be regulated by the accumulation of jasmonates and auxins in roots. Nevertheless, the mechanisms of the interaction of these hormones in the regulation of the growth response to salinity are still not clear enough. Their better understanding depends on the study of the distribution of jasmonates and auxins between root cells. This was achieved with the help of immunolocalization of auxin (indoleacetic acid) and jasmonates on the root sections of pea plants. Salinity inhibited root elongation and decreased the size of the meristem zone and the length of cells in the elongation zone. Immunofluorescence based on the use of appropriate, specific antibodies that recognize auxins and jasmonates revealed an increased abundance of both hormones in the meristem zone. The obtained data suggests the participation of either auxins or jasmonates in the inhibition of cell division, which leads to a decrease in the size of the meristem zone. The level of only auxin and not jasmonate increased in the elongation zone. However, since some literature evidence argues against inhibition of root cell division by auxins, while jasmonates have been shown to inhibit this process, we came to the conclusion that elevated jasmonate is a more likely candidate for inhibiting root meristem activity under salinity conditions. Data suggests that auxins, not jasmonates, reduce cell size in the elongation zone of salt-stressed plants, a suggestion supported by the known ability of auxins to inhibit root cell elongation.

## 1. Introduction

High concentrations of salts in the soil adversely affect the growth and yield of crop plants by decreasing the availability of soil moisture and due to the toxicity effects of NaCl concentration [1]. The area of saline soils is expanding due to global climate change, manifested in increased temperatures and the intensity of droughts. [2]. Irrigation of crops, which is increasingly used in agriculture, also contributes to salinization. Yield losses from salinity can be reduced by increasing the salt tolerance of plants. Therefore, mechanisms of plant adaptation to salinity attract the close attention of researchers. Until recently, the processes that occur in the aerial part of salt-stressed plants were mainly studied [3,4]. However, recently, a number of works have appeared in which more attention has been paid to the processes occurring in plant roots under salinity [5,6,7]. It is shown that changes in the architecture of roots play an important role in the adaptation of plants to salinity [6]. Due to the heterogeneity of the soil, salts are unevenly distributed, and inhibition of the growth of roots that reach saline areas of the soil reduces their contact with a high concentration of soil solution. In addition, growth inhibition releases resources for the realization of protective mechanisms [4], such as the formation of apoplastic barriers that reduce the penetration of toxic ions [8] and the activation of sodium and chloride carriers that prevent the accumulation of toxic ions in plants [7].

Phytohormones play an important role in the adaptation of plants to environmental conditions, including salinity [4]. Special attention has been paid to jasmonic acid (JA) and jasmonates. These hormones trigger processes that provide not only protection for plants from pathogens [9], but also an increase in resistance to abiotic stresses [10]. There is evidence of the ability of jasmonates to increase salt tolerance in plants [11]. Thus, salt-tolerant cultivars were distinguished by a high content of jasmonates [12]. Activation of the processes that provide resistance to both biotic and abiotic stresses is associated with the ability of JA to balance growth and defense activities [13]. Jasmonates play an important role in the regulation of root growth [14,15]. The ability of these growth regulators to inhibit root elongation is one of their characteristic properties [14], and mutants with impaired JA signaling were identified by their inability to inhibit root elongation under the influence of JA [14].

It has been shown that the influence of jasmonates on root growth depends on their cross-talk with auxins [16]. Auxins not only control the development processes of plants but also regulate their stress responses [17]. These hormones, along with jasmonates, are considered crucial in regulating the adaptive responses of plants to salinity [18]. Treatment of plants with jasmonates up-regulates *YUC* genes responsible for the synthesis of auxins [19]. Jasmonic acid-dependent MYC transcription factors have been shown to bind to certain motifs in the *YUC* promoters to regulate stress responses [20]. It was shown that these genes are expressed in roots [21,22], and root elongation was not inhibited by jasmonates in the *YUC* gene mutant [19]. Although many researchers have noted the positive effect of auxins on root branching, high levels of auxins slow elongation of the main roots [23,24] (and references therein). Therefore, the JA-induced increase in the level of auxins may be responsible for the inhibition of root elongation by this hormone. At the same time, it is important to study the distribution of both JA and auxins between cells of root tissues for a clearer understanding of the mechanisms regulating root growth under the influence of these hormones [5]. Such studies have been performed using transgenic plants transformed with constructions containing, for example, the *GUS* gene under the control of promoters sensitive to either JA or auxins. These constructs were used to study the distribution of each of these hormones separately in the cells along the root tip [25,26]. However, it is of interest to compare their distribution between the cells of the root tip in one experiment. This can be performed using the method of immunohistochemical localization using appropriate antibodies. Therefore, we compared the distribution of these hormones in barley roots using antibodies to auxins and ABA [27]. It is important to note that previously, the assessment of the content of JA and auxins under salinity conditions was carried out mainly on transgenic Arabidopsis plants, which are the most easily transformable. The use of immunohistochemical localization makes it possible to analyze the distribution of hormones in any plant species, which is important for deepening our understanding of hormonal regulation of the root responses to salinity in plant species other than Arabidopsis.

The purpose of the work was to reveal the relationship between hormonal and growth reactions in the roots of salt-stressed plants by studying the effects of salinity on the level and distribution of auxins and jasmonates in the apices of pea roots. To achieve this goal, we used immunohistochemical localization with antibodies to these hormones. The choice of peas was dictated by the fact that we have previously obtained data on the localization of abscisic acid in pea roots [28].

## 2. Results

Sodium chloride inhibits elongation of the main root of pea plants (Figure 1a,b). The effect of salinity on root growth was stronger at higher salt concentrations. While 50 mM salinity reduced root length by 5%, the roots of plants grown in a 75 mM NaCl solution were 10% shorter than in the control (plants grown without NaCl). The addition of salt to the nutrient solution did not affect the number of lateral roots.

Salinity reduced the size of the meristem zone, which contains small dividing cells. The meristem size decreased by 16% at 50 mM NaCl and by 23% at 75 mM NaCl relative to control plants that grew without salt (Table 1). The boundary region between the meristematic and elongation zones (between dividing and expanding cells) was detected by the increase in cell size as a result of their transition from division to extension.

The average cell size in the outer layers of the root cortex decreased under the influence of 50 mM and 75 mM NaCl by 10 and 17%, respectively, compared with the control.

Immunohistochemical staining of longitudinal sections of the root tips of control plants with antibodies to jasmonic acid revealed fluorescence in the meristem region, which decreased with the distance from the root tip (Figure 2a). In the region of the developing central cylinder, the staining was weaker than in the developing cortex. The uneven distribution of fluorescence intensity reflected the difference in abundance of jasmonates in the cells of different root zones. The setup at 6× magnification in Figure 3 compared to Figure 2 revealed intense nuclear fluorescence in cell images (Figure 3a), suggesting the presence of jasmonates in these cellular compartments.

The fluorescence of the nuclei was color-coded in blue, indicating its greater intensity relative to other organelles. Thus, the results of immunolocalization of jasmonates revealed their presence in the cells of the root tips and their nuclei, their higher level in the cortex compared to the central cylinder, and their decrease with distance from the root tips.

The presence of sodium chloride in the nutrient solution at a concentration of 50 mM increased the fluorescence intensity compared to the control, which was especially noticeable in the area of the developing central cylinder formed in the meristem, which indicates an increase in the level of jasmonates in the root tips under the influence of salinity (Figure 2b). The fluorescence of cell nuclei under salinity was also higher than that of the control roots (Figure 3b).

Salt stress induced by 75 mM NaCl increased the fluorescence intensity on longitudinal sections of roots treated with antibodies against jasmonates, even more than in the case of 50 mM sodium chloride. (Figure 2c). The fluorescence of many cells was encoded with a blue color, which indicates its high intensity and, accordingly, a high level of jasmonates. However, as in the control, in plants treated with 75 mM NaCl, fluorescence decreased with distance from the root tip. At a distance of 2–2.5 mm from the root tip, similar fluorescence of NaCl-treated and control plants indicated that the content of jasmonates did not differ from the control (Figure 4) in this root zone.

Immunohistochemical staining using antibodies to IAA revealed a pattern somewhat similar to the results of jasmonate immunolocalization. As in the case of antibodies to jasmonates, immunolocalization of auxins with antibodies to IAA revealed a more intense fluorescence at the root tip and a decrease with distance from the tip (Figure 5), indicating a decline in auxin content with increasing distance from the root tip.

At the same time, there were also differences (Figure 5). In contrast to the immunolocalization of jasmonates, more intense fluorescence was observed in the endoderm region both in the control and under saline conditions when antibodies to IAA were used. In plants grown in a solution of 50 mM NaCl, an increased fluorescence, encoded in blue, was observed in the area of the elongation zone of the epidermis, which indicates a higher level of fluorescence and, accordingly, an increased content of auxins in these cells (Figure 5b). The number of blue-encoded cells increased at a higher (75 mM) NaCl concentration, indicating an increase in the level of auxins in these cells.

It Is clearly seen on the cross section of the roots that the content of auxins increased under the influence of 75 mM NaCl in the cortex of the elongation zone (the fluorescence of the cells of this zone was color-coded in blue under high levels of salinity, in contrast to the fluorescence of the cells of the central cylinder and cortex of control plants, which was color-coded in green) (Figure 6).

## 3. Discussion

Adding sodium chloride to the plant nutrient solution inhibited the elongation of pea roots, and the effect increased with higher salt concentrations. Our results are consistent with numerous pieces of literature data on the ability of salinity to inhibit root elongation [6,29]. We found no effect of salinity on root branching; the control and salt-stressed plants did not differ in the number of lateral roots. Information about the effect of sodium chloride on root branching is rather contradictory. Some reports show an increase in the number of lateral roots under the influence of long-term mild salinity [30], while others demonstrate inhibition of lateral root initiation under high concentrations of sodium chloride [31,32]. Obviously, the pattern of the influence of salinity on root branching depends on the concentration of salt, stress duration, and plant growth conditions. In our experiments, the pea plants were grown in hydroponic culture, in contrast to the experiments with Arabidopsis, which were mainly carried out on an agar-containing Murashige-Skoog medium.

Inhibition of root elongation coincided with an increase in the level of jasmonates in the root tips, which was revealed by immunolocalization of these hormones using specific antibodies. It was suggested that, as in the case of ABA [33], the accumulation of jasmonates under salinity is due to a decrease in the availability of water and a loss of turgence [25]. Experiments have shown that salinity increases the expression of the genes controlling the synthesis of jasmonates [25]. The role of jasmonates in the regulation of root growth under salinity is actively discussed [13,14]. A causal relationship between jasmonate accumulation and inhibition of root elongation is supported by the failure of exogenous jasmonates to inhibit root elongation in plants with impaired jasmonate signaling, accompanied by a lack of inhibition of root elongation by salinity [14]. Our data on the accumulation of jasmonates in the root meristem zone are consistent with the data obtained using a β-glucuronidase (*GUS*) reporter gene system sensing the presence of jasmonates [25]. These experiments revealed activation of jasmonate signaling under the influence of salinity in the root meristem zone.

The size of the root meristem zone decreased under the influence of salinity, which can be related to the increase in the level of jasmonates since these hormones are known to negatively affect cell proliferation in roots and leaves [34,35]. Our results, indicating a decrease in the size of the meristem zone due to jasmonates in salt-stressed plants, correspond to the literature data on the effect of either salinity [31,36] or exogenous jasmonates [16] on this parameter. Inhibition of root growth under the influence of salinity can provide the release of resources necessary for the implementation of plant protection from adverse environmental factors. The involvement of jasmonates in this process reflects the hormone’s ability to balance growth and stress responses [11,13].

It is assumed that jasmonates act as a core signal in the phytohormone signaling network. The inhibition of root growth by jasmonates was shown to depend on auxins [14]. Analysis of the transcriptomics of Arabidopsis plants and quantitative determination of auxins and intermediate products of IAA biosynthesis under salinity showed that auxin levels rose due to an increase in IAA biosynthesis [37]. In our experiments with pea plants, we found an accumulation of auxins in the tips of the roots. These results are consistent with those obtained with a reporter construct containing the *GUS* gene under the control of the auxin-sensitive *DR5* promoter [31]. As in our experiments with the immunolocalization of auxins in the roots of pea plants, transgenic Arabidopsis plants showed an increase in *GUS* expression in the meristem and root cap of plants subjected to salt stress, which indicates an increase in the level of auxins in these root zones under the influence of salinity. However, it should be noted that in these experiments with transgenic Arabidopsis plants, these results were observed at 150 mM NaCl, while in our experiments with peas, we used 75 mM NaCl. The accumulation of auxins in the tips of pea roots coincided with the accumulation of jasmonates in them. Our results are consistent with the data on the ability of jasmonates to increase expression of *YUC* genes responsible for the biosynthesis of auxins [19,20]. This proposal by Hetrich et al. was based on data showing that *YUC* knockout mutants had a reduced ability to respond to exogenous jasmonates. Interestingly, the distribution of auxins in the roots of pea plants under salinity resembles the pattern of gene expression that controls the synthesis of auxins [38]. In this cited work, the expression of the genes at the cellular level was assessed using a construct in which the *GUS* reporter gene was placed under the control of the *YUC* gene promoter. As in the case of auxin immunolocalization in experiments with salt-stressed pea plants, the expression of various genes of the *YUC* family was found in the root tip and in the zone of elongation, as well as at the border between the central cylinder and cortex. Although the data obtained with Arabidopsis cannot be extrapolated with certainty to pea plants, similarity in the distribution of auxins and *YUC* gene expression suggests that the increase in the level of auxins in the roots under salinity could result from their increased synthesis under the influence of jasmonates.

At the same time, it is difficult to unequivocally link the decrease in meristem size induced by salt with the accumulation of auxins in the root tips, since some evidence suggests that these hormones can positively influence root cell proliferation and thus meristem size. Thus, Liu et al. hypothesized that salt stress reduces the size of the root meristem by suppressing expression of the *PINFORMED* (*PIN*) genes encoding auxin transporters, thus reducing the level of auxin (i.e., root tip proliferation decreases with a decrease rather than an increase in auxin levels) [32]. Chapman et al. argue that cell division in the proximal root meristem is maintained by the presence of a high concentration of auxins [39]. These data do not allow us to explain the decrease in the size of the meristem zone, which we found in the roots of pea plants, by the accumulation of auxins in the root tips under salinity. It is most likely that we can assume a direct inhibitory effect of jasmonates on the activity and size of the meristem of salt-stressed pea roots.

At the same time, the decrease in cell size observed by us in the elongation zone of pea roots under salinity cannot be explained by the direct action of jasmonates, since the abundance of these hormones in the cells of the root elongation zone of salt-stressed plants was similar to the control. In this case, it is logical to involve auxins to explain the changes in pea root elongation under salinity, since these hormones decrease the cell size in the elongation zone of the root [23,40]. This makes it possible to associate the decrease in the rate of root elongation with the accumulation of auxins in the cells of the elongation zone, which we observed in pea roots under salinity. The increase in the level of auxins during salinity observed in the cortex of the elongation zone of pea roots is consistent with the data obtained on Arabidopsis with auxin-sensitive reporter constructs [26], which showed that *GUS* expression is mainly detected in the cells of the epidermis of the elongation zone. In the experiments of Smolko et al., a translocation of the auxin signal into the cortex and epidermis of Arabidopsis roots under salinity was also found [26]. Changes in the distribution of auxins between root zones are associated with the activity of auxin transporters [32]. However, it is possible that the accumulation of IAA in the elongation zone may be associated with the activation of the synthesis of this hormone, since some genes from the *YUC* family are expressed in the elongation zone [38], and their expression can be influenced by jasmonates.

## 4. Materials and Methods

### 4.1. Plant Growth and Collecting Plant Material

Studies were carried out on a pea (*Pisum sativum* L.) cultivar, Sakharny 2. Before the start of the experiments, pea seeds were disinfected with a mixture of 96% ethanol and 3% hydrogen peroxide (in a ratio of 3:1) for 10 min as described [28]. Then the seeds were thoroughly washed under running water and left overnight in water to swell under continuously aerated conditions. The next day, the seeds were laid out on the trays, wrapped in wet paper towels, and covered with glass, leaving space for air to enter. For the next two days, the trays with seeds were kept in the dark with water added daily. Two days later, 4-day-old seedlings were transplanted on floating beds with a diameter of 8 cm made of polyethylene foam. There were 8 holes for plant seedlings in each bed. Plants were grown in hydroponic culture on 10% Hoagland–Arnon nutrient solution in 5-liter containers at 400–500 µmol m^−2^s^−1^ PAR (ZN-500 and DNAT400 lamps), 24/18 °C (day/night), and 14-h photoperiod, maintaining aeration of the solution. On the third day of cultivation on a light platform, part of the plants was transplanted into solutions of sodium chloride at a final concentration of 50 mM and 75 mM in a 10% Hoagland–Arnon solution. Control plants were transplanted into a fresh 10% Hoagland–Arnon solution. Subsequently, the replacement of solutions with fresh ones was performed every day.

On the fourth day after the transfer of plants to NaCl solutions, the length of the roots was measured, and root tips were fixed for immunolocalization of jasmonic acid and auxins. The length of cells in the elongation zone was measured on unstained preparations.

### 4.2. Immunolocalization of Jasmonic Acid and Auxins

Root tips with a length of 5 mm were cut with a sharp razor blade in a small amount of water. Root tissues of control and treated plants were fixed in a freshly prepared 4% solution of 1-ethyl-3-(3-dimethylaminopropyl) carbodiimide (Sigma, St. Louis, MO, USA) in phosphate-saline buffer (PBS), pH 7.2 [41,42]. For better penetration of the fixative solution into the tissues, the root pieces were placed under vacuum during the first 30 min of fixation. After infiltration, the fixative solution was replaced with a fresh one and left for 5–6 h at 4 °C. Then post-fixation of plant tissues was carried out in 4% paraformaldehyde solution (Riedel de Haen, Seelze, Germany) and 0.1% glutaraldehyde (Sigma, Steinheim, Germany) prepared in PBS (pH 7.2) for 12 h at 4 °C. The next day, plant tissues were washed in PBS (pH 7.2) twice for 60 min and dehydrated in a series of ethanol dilutions (10%, 20%, 30%, 40%, 50%, 60%, 70%, 80%, 90%, and 96%), keeping in each for 30–40 min at 4 °C. Embedding of tissue samples in JB4 resin (Sigma, St. Louis, MO, USA) was carried out according to the manufacturer’s recommendations in special molds for embedding. After complete solidification of the resin, 1.5-micormeter-thick sections were obtained on a rotary microtome HM 325 (MICROM Laborgerate, Walldorf, Germany). The sections were then transferred to a drop of water on glass slides and dried. Before applying the immune serum containing specific antibodies to the studied phytohormones, a solution of PBS (pH 7.2) with 0.2% gelatin and 0.05% Tween 20 (PGT) in a volume of 50 µL was dripped onto each section, then covered with parafilm and kept at room temperature for 30 min in a humidity chamber. A humidity chamber was created in Petri dishes by placing filter paper moistened with distilled water on the bottom. After a single rinse in distilled water, slides were dried, and then rabbit polyclonal antibodies to jasmonic acid (1:1000) (Agrisera, Vannas, Sweden) or auxins (1:50) [20] were applied to each section in a volume of 20 µL. Dilutions of immune sera were prepared in PGT solution according to the attached instructions. Slides covered with parafilm were placed in a humidity chamber and kept at 4 °C overnight. To confirm the immune specificity of antibodies, serum containing no antibodies to the studied hormones was used. The next day, parafilm was removed from the sections, and the slides were placed in glass containers for washing in a solution of PBS (pH 7.2) containing Tween 20 (0.05%) (three times for 10 min). To detect the localization of phytohormones on histological preparations, goat anti-rabbit secondary antibodies labeled with the Alexa555 fluorescent dye (Invitrogen, Rockford, IL, USA) were applied to the sections. The dilution of the Alexa555 (1:500, according to the instructions) was prepared on PGT, applied in a volume of 20 μL to each section, and covered with parafilm. To incubate sections with secondary antibodies, slides were placed in a humid chamber and kept at 37 °C for three hours. At the final stage, the slides were washed five times for 10 min in a PBS solution, rinsed with water, and a mixture of glycerol and gelatin was applied to the sections, which were immediately covered with coverslips. A mixture of glycerin and gelatin was prepared in the following ratio: 0.5 g of gelatin and 3.5 mL of glycerol in 3 mL of distilled water. Fluorescence images were obtained using an Olympus FV3000 Fluoview (FV31-HSD) confocal laser scanning microscope (Olympus, Tokyo, Japan) with an excitation laser wavelength of 561 nm. Fluorescence emission was detected at 568 nm in integration frame mode for imaging with a count of 2. The difference in intensity of staining was displayed by a color-coded heatmap. In order to be able to compare images and measure fluorescence, standard laser power and signal detection settings were used in all cases. They differed for two hormones but were the same for control and NaCl treatments. Seven to 15 root sections were used to assess the localization of hormones. Averaged fluorescence per one pixel registered with the confocal microscope was estimated with ImageJ version 1.53 software (National Institutes of Health, Bethesda, MD, USA, https://imagej.nih.gov/).

### 4.3. Measurement of the Meristem Zone and Cell Length

The size of the meristem zone and the length of cells in the outer layers of cortex in the root elongation zone were measured on unstained preparations of longitudinal sections of the roots of the control and salt-stressed plants. The histological preparations were analyzed using a light microscope (Carl Zeiss, Jena, Germany) equipped with a digital camera (AxioCam MRc5, Carl Zeiss, Jena, Germany). Cell measurements were made using the AxioVision 4 software.

### 4.4. Statistics

The experiments were repeated three times. The length of the primary roots of 100 plants was measured, and the size of the meristem zone was measured in 25 roots, while the size of 100 cells in the elongation zone of each variant was estimated. The data were processed using the Statistica version 10 software (Statsoft, Moscow, Russia). In figures and tables, data are presented as mean ± standard error (s.e.). The significance of differences was assessed by ANOVA, followed by Duncan’s test (*p* < 0.05) or *t*-test.

## 5. Conclusions

An analysis of our results and literature data indicates a relationship between the accumulation of jasmonates and auxins in the roots and inhibition of elongation of pea roots under saline conditions due to a decrease in the size of both the meristem zone and cells of the cortex in the elongation zone (Figure 7).

The study of the distribution of phytohormones in different root zones in the present experiments and the analysis of the relevant literature indicating their influence on the processes of division and elongation of the roots suggest that the reduction in the size of the pea root meristem revealed during salinity is most likely associated with the accumulation of jasmonates in the root tips, and a decrease in the length of cells in the elongation zone is associated with the accumulation of auxins in the cortex of the root elongation zone. Changes in the content of auxins during salinity are likely to be the result of an increase in the expression level of genes that control their synthesis, and jasmonates can be involved in such an increase in the expression of these genes. Testing this hypothesis should be the goal of further research.

## Figures and Tables

**Figure 1 ijms-24-15148-f001:**
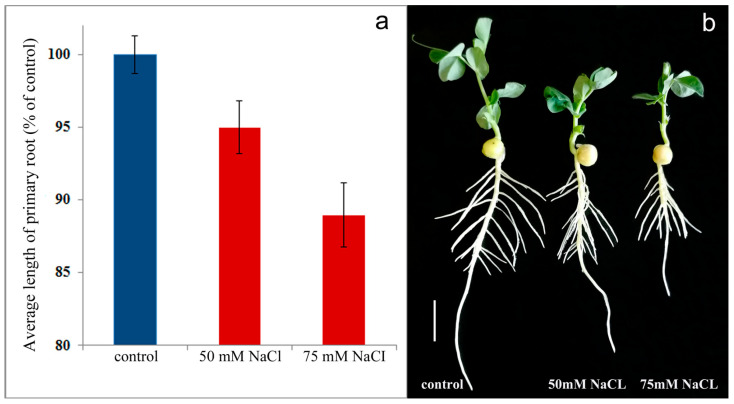
The effect of salt stress on the average length of primary roots of 11-days-old pea plants, measured 4 days after addition of sodium chloride at concentrations of 50 mM and 75 mM to the nutrient solution (*n* = 100) (blue—control plants, red—plants under salt stress). The data were processed using the Statistica version 10 software (Statsoft, Moscow, Russia). The results are presented as mean ± standard error (s.e.). The significance of differences was assessed by ANOVA, followed by Duncan’s test (*p* < 0.05). Significantly different mean values represented by the bars are marked by different letters (*p* ≤ 0.05, Duncan’s test) (**a**). Image of control and salt-stressed plants (**b**). The scale bar is 2 cm.

**Figure 2 ijms-24-15148-f002:**
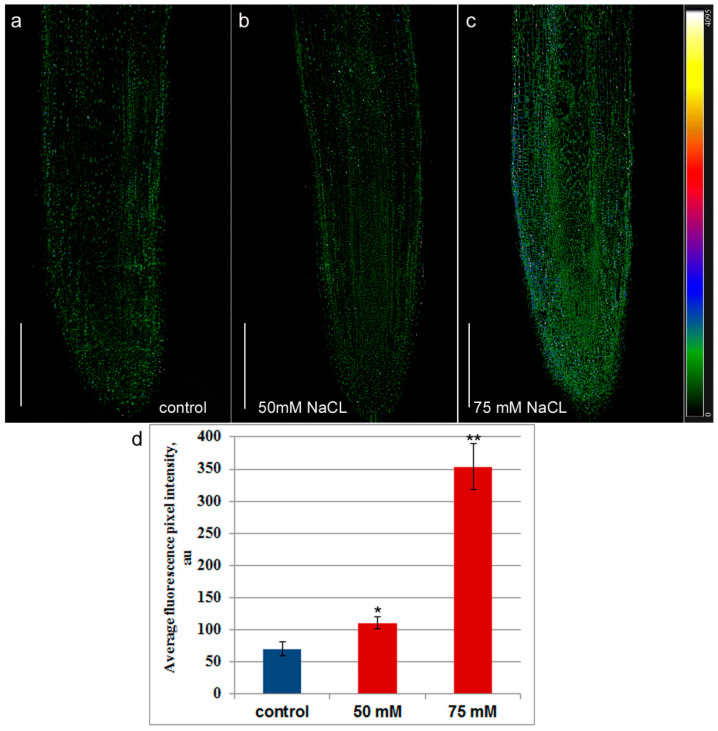
Immunohistochemical staining of jasmonates on the longitudinal root sections of 11-day-old pea plants grown without NaCl (**a**) and in solution with 50 mM (**b**) and 75 mM NaCl (**c**). The intensity of fluorescence of the second antibody against rabbit immunoglobulins is color-coded; a green color corresponds to lower fluorescence, while blue and red colors reflect a gradual increase in fluorescence. The scale bar is 500 µm. (**d**) Fluorescence intensity in the region of the developing central cylinder of the meristem. The averaged fluorescence per one pixel registered with the confocal microscope was estimated with ImageJ software. One and two asterisks indicate means significantly different from the control treatment at *p* ≤ 0.05 and *p* ≤ 0.01, respectively (*t*-test).

**Figure 3 ijms-24-15148-f003:**
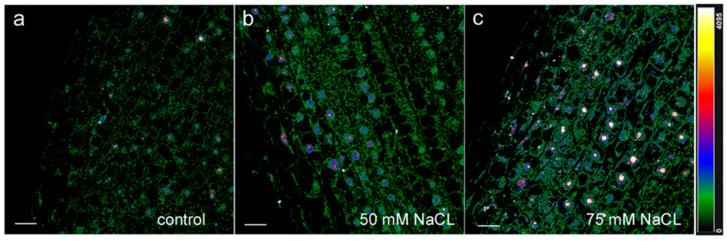
An enlarged image of immunohistochemical staining with rabbit antibodies against jasmonates of fragments of longitudinal root sections located at 500 µm (0.5 mm) from the root apex of 11-day-old pea plants grown without NaCl (**a**) and in solution with 50 mM (**b**) and 75 mM NaCl (**c**) Images are presented at a greater magnification compared to Figure 2. The intensity of fluorescence of the second antibody against rabbit immunoglobulins is color-coded; a green color corresponds to lower fluorescence, while blue and red colors reflect a gradual increase in fluorescence. The scale bar is 20 µm.

**Figure 4 ijms-24-15148-f004:**
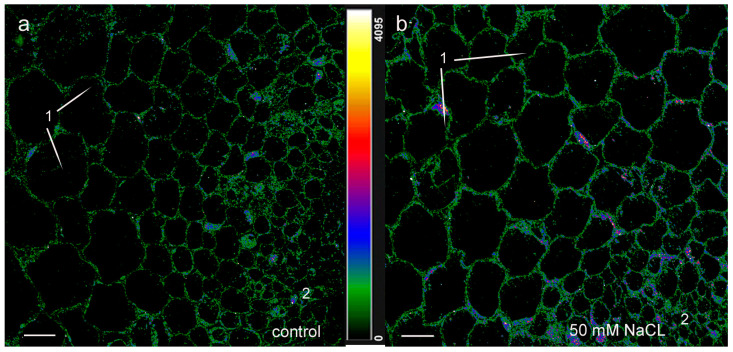
Immunohistochemical staining with rabbit antibodies against jasmonates of root cross sections cut at 2–2.5 mm from the root apex of 11-day-old pea plants grown without NaCl (**a**) and in solution with 75 mM NaCl (**b**). Images are presented at a greater magnification compared to Figure 2. The intensity of fluorescence of the second antibody against rabbit immunoglobulins is color-coded; a green color corresponds to lower fluorescence, while blue and red colors reflect a gradual increase in fluorescence. The scale bar is 20 µm. 1—cortex and 2—central cylinder.

**Figure 5 ijms-24-15148-f005:**
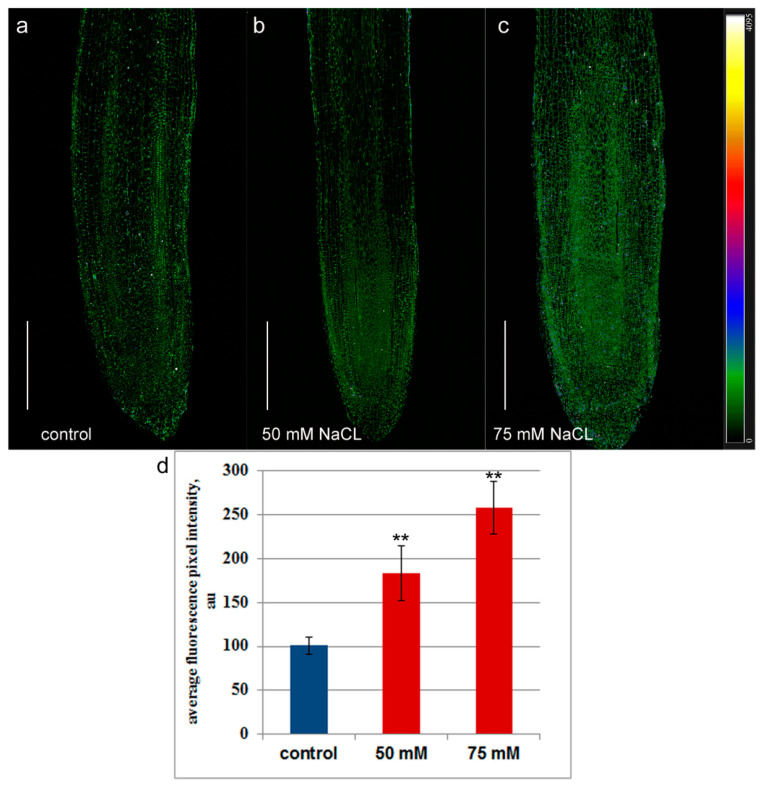
Immunohistochemical staining using rabbit antibodies against indole-acetic acid (IAA) of the longitudinal root sections of 11-day-old pea plants grown without NaCl (**a**) and in solution with 50 mM (**b**) and 75 mM NaCl (**c**). The intensity of fluorescence of the second antibody against rabbit immunoglobulins is color-coded; a green color corresponds to lower fluorescence, while blue and red colors reflect a gradual increase in fluorescence. The scale bar is 500 µm. (**d**) Fluorescence intensity in the region of the meristem. The averaged fluorescence per one pixel registered with the confocal microscope was estimated with ImageJ software. Asterisks indicate means that are significantly different from the control treatment at *p* ≤ 0.01 (*t*-test).

**Figure 6 ijms-24-15148-f006:**
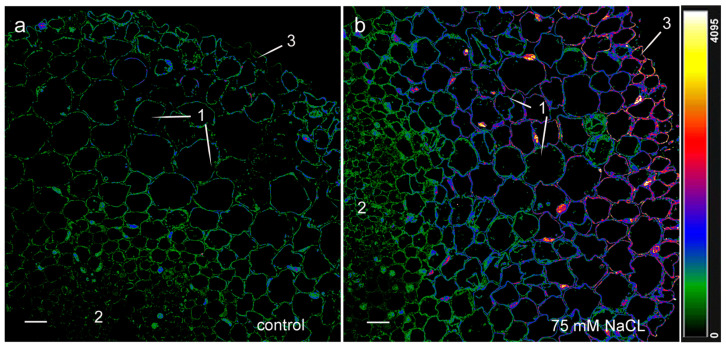
Immunohistochemical staining with rabbit antibodies against IAA of root cross sections cut at 2–2.5 mm from the root apex of 11-day-old pea plants grown without NaCl (**a**) and in solution with 75 mM NaCl (**b**). Images are presented at greater magnification. The intensity of fluorescence of the second antibody against rabbit immunoglobulins is color-coded; a green color corresponds to lower fluorescence, while blue and red colors reflect a gradual increase in fluorescence. The scale bar is 20 µm. 1—cortex; 2—central cylinder; and 3—exodermis.

**Figure 7 ijms-24-15148-f007:**
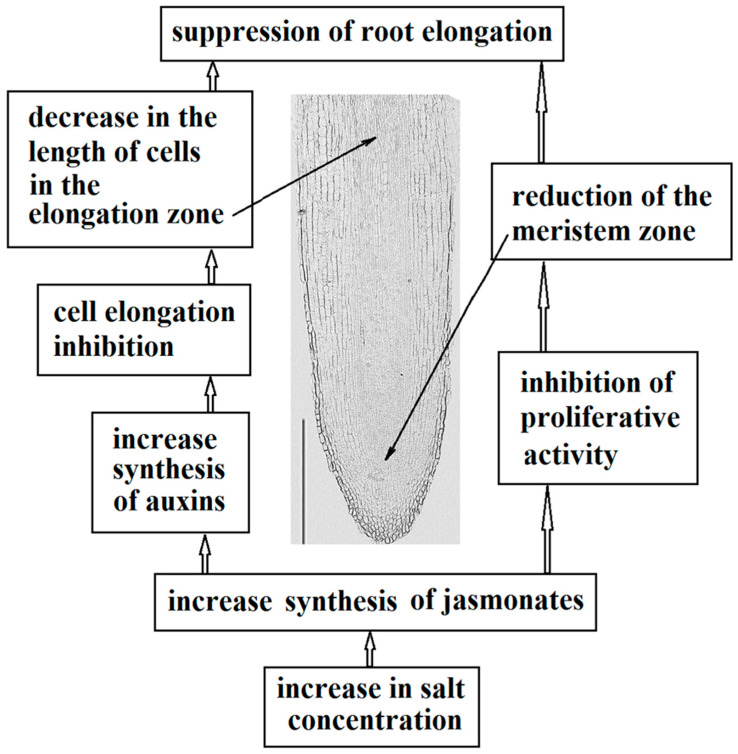
Scheme of salinity action on elongation of pea main roots due to salt-induced changes in the content of jasmonates and auxins, summarizing the results of the present experiments and literature data. The scale bar is 500 µm.

**Table 1 ijms-24-15148-t001:** The effect of salinity on the size of the meristem zone (*n* = 25) and average cell length of the outer layers of root cortex cells in the elongation zone (*n* = 100) of 11-day-old pea plants grown at 400–500 µmol m^−2^s^−1^ PAR, 24/18 °C (day/night), and 14-h photoperiod was measured 4 days after adding 50 and 75 mM sodium chloride to the nutrient solution. Significantly different values are denoted by different letters (*p* ≤ 0.05, Duncan test).

NaCl Stress	Meristem Size, µm	Mean Length of Cells in Elongating Root Zone, µm
0 mM NaCl	1066 ± 86 ^a^	60 ± 2 ^a^
50 mM NaCl	899 ± 29 ^b^	54 ± 1 ^b^
75 mM NaCl	822 ± 45 ^b^	50 ± 3 ^b^

## Data Availability

Data is contained within the article.

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
