# Peer review of "Immunolocalization of Jasmonates and Auxins in Pea Roots in Connection with Inhibition of Root Growth under Salinity Conditions"

_ijms, 2023, doi:10.3390/ijms242015148_

Round 1

Reviewer 1 Report

1.I will suggest authors to modify the abstract part including the major findings of the current study with 2 lines of conclusion of the current study.

2. I will encourage to include quantification data for these phytohormones.

3. Authors can include transcriptomic study of a few genes that are majorly involved in the regulation of hormonal synthesis in plants.

4. I will suggest including 1 figure of the whole root in different treatments.

Please see pdf for more corrections

Author Response

We are most grateful to the respected reviewer for careful reading of our article and valuable comments. We have done our best to follow them

1.I will suggest authors to modify the abstract part including the major findings of the current study with 2 lines of conclusion of the current study.

Response: Abstract was modified by including more findings: “Salinity inhibited root elongation, decreased the size of meristem zone and the length of cells in the elongation zone. Immunofluorescence based on the use of appropriate specific antibodies that recognize auxins and jasmonates revealed increased abundance of both hormones in the meristem zone. The obtained data suggests participation of either auxins or jasmonates in the inhibition of cell division, which leads to a decrease in the size of the meristem zone. The level of only auxin, and not jasmonate, increased in the elongation zone.”

To specify conclusion we included several lines: “since some literature evidence argues against inhibition of root cell division by auxins, while jasmonates have been shown to inhibit this process, WE CAME TO THE CONCLUSION that elevated jasmonate is a more likely candidate for inhibiting root meristem activity under salinity conditions. Data suggests that auxins, not jasmonates, reduce cell size in the elongation zone of salt-stressed plants, a suggestion supported by the known ability of auxins to inhibit root cell elongation”

  1. I will encourage to include quantification data for these phytohormones.

Response: To follow this recommendation of respected reviewer we added quantification of immunofluorescence related to hormone abudance with the help of ImajeJ software. We added description of this approach to M & M section: “Averaged fluorescence per one pixel registered with the confocal microscope were estimated with ImageJ software (National Institutes of Health, Bethesda, MD, USA, https:// imagej. nih. gov/ ij).” The results are presented in Fig. 2d and 5d.

  1. Authors can include transcriptomic study of a few genes that are majorly involved in the regulation of hormonal synthesis in plants.

Response: Our suggestion about effect of jasmonates on expression of YUC genes responsible for auxin synthesis is based on corresponding literature data. We understand that study of expression of the gene in our research would be useful and we are grateful to the reviewer for this recommendation. However we were given only 10 days for revision of our article and this is not sufficient for performing such experiments. Furthermore, possibility to continue our work depends on publication of the present results this year. So at present, in accordance with this remark, we just added at the end of Conclusion (after the sentence telling “Changes in the content of auxins during salinity are likely to be the result of an increase in the expression level of genes that control their synthesis, and jasmonates can be involved in such an increase in the expression of these genes”) that “Testing this hypothesis should be the goal of further research”.

  1. I will suggest including 1 figure of the whole root in different treatments.

Response: in the revised article, image of the plants is presented in Figure 1b.

Please see pdf for more corrections

  1. Figure 1. Include the statistical test used in the figure legends as well as the description about alphabet on the bars.

Response: According to this recommendation we added to the figure legend that “The data were processed using the Statistica version 10 software (Statsoft, Moscow, Russia). The results are presented as mean ± standard error (s.e.). The significance of differences was assessed by ANOVA, followed by Duncan’s test (p < 0.05).”

Description of alphabet on the bars is explained in the following sentence: “Significantly different mean values represented by the bars are marked by different letters (p≤0.05, Duncan's test).”

  1. In the comments to the pdf file of our article respected reviewer recommended to add a reference to the description of disinfection, which was done.
  2. As indicated above, in the revised article, image of the plants is presented in Figure 1b.

Reviewer 2 Report

Dear Authors 

I appreciate the idea and effort. There are some minor corrections are needed.

The title of manuscript should be changed as the work not clearly showing the Role of Jasmonic Acid and Auxins in the Regulation of Root Elongation. 

Abstract; Some more information about results are needed.

Introduction line 25; I don't think soil is a solution. 

M&M

Please provide biological and technical replicates.

Line 299; "Root tips 5 mm long were cut", it seems grammatically wrong, maybe Root tips with 5 mm length?

Line 300; "Root tissues of control and experimental plants were fixed", control plants are also part of experiment maybe it would be better to write control and treated plants.

Best wishes

English quality is good some minor correction are needed.

Author Response

We are most grateful to the respected reviewer for careful reading of our article and valuable comments. We have done our best to follow them

  1. The title of manuscript should be changed as the work not clearly showing the Role of Jasmonic Acid and Auxins in the Regulation of Root Elongation. 

Response: The title of the article has been changed: “Immunolocalization of Jasmonates and Auxins In Pea Roots in Connection with Inhibition of Root Growth under Salinity Conditions”

  1. Abstract; Some more information about results are needed.

Response: More information was added: “Salinity inhibited root elongation, decreased the size of meristem zone and the length of cells in the elongation zone. Immunofluorescence based on the use of appropriate specific antibodies that recognize auxins and jasmonates revealed increased abundance of both hormones in the meristem zone. The obtained data suggests participation of either auxins or jasmonates in the inhibition of cell division, which leads to a decrease in the size of the meristem zone. The level of only auxin, and not jasmonate, increased in the elongation zone”.

  1. Introduction line 25; I don't think soil is a solution. 

Response: “solution” was deleted

M&M

  1. Please provide biological and technical replicates.

Response: In accordance with this remark we added to M & M section that “Experiments were repeated three times. The length of primary roots of 100 plants was measured and the size of meristem zone was measured in 25 roots, while the size of 100 cells in the elongation zone of each variant was estimated.”

  1. Line 299; "Root tips 5 mm long were cut", it seems grammatically wrong, maybe Root tips with 5 mm length?

Response: This was changed according to recommendation of respected reviewer.

  1. Line 300; "Root tissues of control and experimental plants were fixed", control plants are also part of experiment maybe it would be better to write control and treated plants.

Response: The phase was modified according to recommendation of reviewer

Reviewer 3 Report

The manuscript titled “The Role of Jasmonic Acid and Auxins in the Regulation of Root Elongation in Pea Plants under Salinity” reports an interesting work and focused on the effect of salinity on the level and distribution of indoleacetic (IAA) and jasmonic acids  (JA) in pea roots using immunohistochemical localization with antibodies to these hormones. This is a well-written article and I anticipate that the manuscript should be of great interest to the researchers working on plant hormones and modulation of hormones under stressful conditions. I include my comments below, most of them are suggestion to improve the overall quality for publication. I considered the manuscript suitable for publication subject to following improvements.

Abstract:

1.      The authors elaborated abstract in a good way. However, some prominent results should be added to improve this section, and add a concluding remark at the end of abstract section.

2.      It is suggested to revise and elaborate the statement “Salinity inhibited root elongation, which was accompanied by accumulation of both hormones in the meristem zone, while the content of only auxin, and not jasmonate, increased in the elongation zone”.

3.      Rearrange the keywords alphabetically.

Introduction:

4.      Revise the statement and cite appropriately, Line 27-29: High concentration of salts in the soil solution adversely affects the development of plants and their productivity. Saline soils are spread worldwide, and their areas are constantly expanding due to the ever greater aridity of the climate, which increases evaporation of water, while salts coming with water from the depth of the soil, remain in the upper soil layer.

5.      Line 33-34, revise the statement as follow; “However, recently several articles reported the processes occurring in the roots of plants under salt stress”.

6.      It is suggested to divide the second paragraph of introduction section into two paragraphs, and add information regarding Indoleacetic acids (IAA). Moreover, both paragraph should be revised and be added with recent literature in line with the study.

7.      The objectives of the study should be revised and readers friendly.

Results

8.      In results section, Table 1. Plant growing conditions should be different from salt stress. In plant growing conditions, the author should add light intensity, humidity, temperature. While the column should be labelled as “NaCl stress”. You can also denote “control” as 0mM NaCl.

9.      Revise the caption of figure 2. Immunohistochemical staining with rabbit antibodies against jasmonates of the longitudinal root sections of 11-days old pea plants grown in solution with (a) 0mM NaCl, (b) 50 mM NaCl (c) 75 mM NaCl.

10.  Line 107: add space (Figure 2a). In the…

11.  In figure 2. Intensity of fluorescence of the second antibodies against rabbit immunoglobulins is color-coded, green color corresponding to lower fluorescence, while blue and red coloring reflect gradual increase in fluorescence. However, the fluorescence is not equally distributed, can you add the reason?

12.  Line 170: Images are presented at a greater magnification. This should be self-explanatory. It is suggested to add some details.

13.   Figure 7, should be moved to introduction part.

Discussion

14.  Should be more robust by adding recent references and compare your studies.

Materials and Methods

15.  Well written

16.  It is suggested to divide the section into two paragraphs “Immunolocalization of jasmonic acid and auxins”.

Author Response

Третий

We are most grateful to the respected reviewer for careful reading of our article and valuable comments. We have done our best to follow them

The manuscript titled “The Role of Jasmonic Acid and Auxins in the Regulation of Root Elongation in Pea Plants under Salinity” reports an interesting work and focused on the effect of salinity on the level and distribution of indoleacetic (IAA) and jasmonic acids  (JA) in pea roots using immunohistochemical localization with antibodies to these hormones. This is a well-written article and I anticipate that the manuscript should be of great interest to the researchers working on plant hormones and modulation of hormones under stressful conditions. I include my comments below, most of them are suggestion to improve the overall quality for publication. I considered the manuscript suitable for publication subject to following improvements.

Abstract:

  1. The authors elaborated abstract in a good way. However, some prominent results should be added to improve this section, and add a concluding remark at the end of abstract section.

Response: The abstract was modified. Changes are as follows: Salinity inhibited root elongation, decreased the size of meristem zone and the length of cells in the elongation zone. Immunofluorescence based on the use of appropriate specific antibodies that recognize auxins and jasmonates revealed increased abundance of both hormones in the meristem zone. The obtained data suggests participation of either auxins or jasmonates in the inhibition of cell division, which leads to a decrease in the size of the meristem zone. The level of only auxin, and not jasmonate, increased in the elongation zone. However, since some literature evidence argues against inhibition of root cell division by auxins, while jasmonates have been shown to inhibit this process, we came to the conclusion that elevated jasmonate is a more likely candidate for inhibiting root meristem activity under salinity conditions. Data suggests that auxins, not jasmonates, reduce cell size in the elongation zone of salt-stressed plants, a suggestion supported by the known ability of auxins to inhibit root cell elongation.”

  1. It is suggested to revise and elaborate the statement “Salinity inhibited root elongation, which was accompanied by accumulation of both hormones in the meristem zone, while the content of only auxin, and not jasmonate, increased in the elongation zone”.

These sentences have been changed (see above)

  1. Rearrange the keywords alphabetically.

Response: Succession of key words was arranged alphabetically: auxin; immunolocalization; jasmonic acid; root length; root meristematic zone; Pisum sativum L.

Introduction:

  1. Revise the statement and cite appropriately, Line 27-29: High concentration of salts in the soil solution adversely affects the development of plants and their productivity. Saline soils are spread worldwide, and their areas are constantly expanding due to the ever greater aridity of the climate, which increases evaporation of water, while salts coming with water from the depth of the soil, remain in the upper soil layer.

Response: The statements were modified according to more appropriate citations: High concentration of salts in the soil adversely affects the growth and yield of crop plants by decreasing the availability of soil moisture, and due to the toxicity effects of NaCl concentration [El Sabagh et al. 2021]. The area of saline soils is expanding brought about by global climate change, manifested in increased temperatures and intensity of droughts. [Navarro-Torre et al., 2023]

  1. Line 33-34, revise the statement as follow; “However, recently several articles reported the processes occurring in the roots of plants under salt stress”.

Response: the statement was revised: “However, recently a number of works have appeared in which more attention was paid to the processes occurring in plant roots under salinity.”

  1. It is suggested to divide the second paragraph of introduction section into two paragraphs, and add information regarding Indoleacetic acids (IAA). Moreover, both paragraph should be revised and be added with recent literature in line with the study.

Response: The second paragraph is divided and recent literature added. It is now modified in the following way: “Auxins not only control development processes of plants, but also regulate their stress responses [Abdel Latef et al., 2021]. These hormones, along with jasmonates, are considered crucial in regulating adaptive responses of plants to salinity [Verma et al., 2022]. Treatment of plants with jasmonates up-regulates YUC genes responsible for the synthesis of auxins [15]. Jasmonic acid-dependent MYC transcription factors have been shown to bind to certain motifs in the YUC promoters to regulate stress responses [Pérez-Alonso et al,. 2021]. It was shown that these genes are expressed in roots [16+ Cao et al., 2019], and root elongation was not inhibited by jasmonates in the YUC gene mutant [15]. Although many researchers have noted the positive effect of auxins on root branching, high level of auxins slow elongation of the main roots [17+ Kudoyarova et al., 2023 and references therein].

  1. The objectives of the study should be revised and readers friendly.

Response: Objectives of the study were revised in the following way: “The purpose of the work was to reveal the relationship between hormonal and growth reactions in the roots of salt-stressed plants by studying the effects of salinity on the level and distribution of auxins and jasmonates in the apices of pea roots. To achieve this goal we used immunohistochemical localization with antibodies to these hormones.

Results

  1. In results section, Table 1. Plant growing conditions should be different from salt stress. In plant growing conditions, the author should add light intensity, humidity, temperature. While the column should be labelled as “NaCl stress”. You can also denote “control” as 0mM NaCl.

Response: Table 1 legend was modified by introducing growing conditions. It is now as follows: “The effect of salinity on the size of meristem zone (n=25) and average cell length of the outer layers of root cortex cells in the elongation zone (n=100) of 11-days-old pea plants grown at 400-500 µmol m-2s-1 PAR, 24/18°C (day/night) and 14-hour photoperiod measured 4 days after adding 50 and 75 mM sodium chloride to the nutrient solution. Significantly different values are denoted by different letters (p≤0.05, Duncan test).

Column is now labeled as NaCl stress and “control” substituted with 0 mM NaCl

  1. Revise the caption of figure 2. Immunohistochemical staining with rabbit antibodies against jasmonates of the longitudinal root sections of 11-days old pea plants grown in solution with (a) 0mM NaCl, (b) 50 mM NaCl (c) 75 mM NaCl.

Response: The caption of figure 2 is modified according to recommendation of respected reviewer as follows: “Immunohistochemical staining of jasmonates on the longitudinal root sections of 11-days old pea plants grown in solution with (a) 0 mM NaCl, (b) 50 mM NaCl (c) 75 mM NaCl.”

  1. Line 107: add space (Figure 2a). In the…

Response: Thanks! Space is added

  1. In figure 2. Intensity of fluorescence of the second antibodies against rabbit immunoglobulins is color-coded, green color corresponding to lower fluorescence, while blue and red coloring reflect gradual increase in fluorescence. However, the fluorescence is not equally distributed, can you add the reason?

Response: In accordance with this remark we added that “Uneven distribution of fluorescence intensity reflected the difference in abundance of jasmonates in the cells of different root zones.” We also added similar reasoning of uneven distribution of fluorescence in the case of immunostaining of auxins: “As in the case of antibodies to jasmonates, immunolocalization of auxins with antibodies to IAA revealed a more intense fluorescence at the root tip and its decrease with distance from the tip (Figure 5) indicating a decline in auxin content with increasing distance from the root tip.” And further in the Discussion these results are supported by reference to literature: “Our data on the accumulation of jasmonates in the root meristem zone are consistent with the data obtained using a β-glucuronidase (GUS) reporter gene system sensing the presence of jasmonates”. And further about auxins: “As in our experiments with the immunolocalization of auxins in the roots of pea plants, transgenic Arabidopsis plants showed an increase in GUS expression in the meristem and root cap of plants subjected to salt stress, which indicates an increase in the level of auxins in these root zones under the influence of salinity.”

  1. Line 170: Images are presented at a greater magnification. This should be self-explanatory. It is suggested to add some details.

Response: According to the remark of the respected reviewer, description of the image was modified.  In the figure capture “An enlarged image”was added to “immunohistochemical staining” and it was further specified that“The setup at 6x magnification in Figure 3 compared to Figure 2 revealed intense nuclear fluorescence in cell images (Figure 3a), suggesting the presence of jasmonates in these cellular compartments”.

  1. Figure 7, should be moved to introduction part.

Response: Since this figure summarizes not only literature data, but analysis of our own results we believe that Conclusion is its proper place. However the origin of this figure was obviously not properly explained and we added to the figure legend a phrase clarifying this point: “Scheme of salinity action on elongation of pea main roots due to salt-induced changes in the content of jasmonates and auxins SUMMARIZING RESULTS OF PRESENT EXPERIMENTS AND LITERATURE DATA” (added phrase is in capital letters here)

Discussion

  1. Should be more robust by adding recent references and compare your studies.

Response: We added 6 more references to article published mostly after 2020 and not earlier than in 2015. As a result, in the revised article, 70 % of articles cited in Discussion have been published no earlier than 2015 (mostly after 2020)

Materials and Methods

  1. Well written
  2. It is suggested to divide the section into two paragraphs “Immunolocalization of jasmonic acid and auxins”.

Response: If we divide this section into description of immunolocalization of jasmonic acid and auxins, the two paragraphs will be repetitions of exactly similar procedures differing only in one sentence concerning treatment of sections with primary antibodies either to jasmonates or auxins.

Round 2

Reviewer 3 Report

In the discussion section cite: 10.9787/PBB.2018.6.3.221